# Ocean Acidification Affects the Response of the Coastal Coccolithophore *Pleurochrysis carterae* to Irradiance

**DOI:** 10.3390/biology12091249

**Published:** 2023-09-18

**Authors:** Fengxia Wu, Jia Guo, Haozhen Duan, Tongtong Li, Yanan Wang, Yuntao Wang, Shiqiang Wang, Yuanyuan Feng

**Affiliations:** 1College of Marine and Environment, Tianjin University of Science and Technology, Tianjin 300453, China; 2School of Oceanography, Shanghai Jiao Tong University, Shanghai 200040, China; 3State Key Laboratory of Satellite Ocean Environment Dynamics, Second Institute of Oceanography, Ministry of Natural Resources, Hangzhou 310012, China; 4Shanghai Frontiers Science Center of Polar Science (SCOPS), Shanghai 200030, China

**Keywords:** phytoplankton, coccolithophores, CO_2_, irradiance, photosynthesis, calcification, carbon fixation

## Abstract

**Simple Summary:**

Here, we investigated the response of the coccolithophore *Pleurochrysis carterae (P. carterae*) isolated from the Bohai Sea to ocean acidification under different irradiance levels. A full factorial matrix of two *p*CO_2_ conditions (400 ppm and 800 ppm) and irradiance levels of 50, 200, 500 and 800 μmol photons m^−2^ s^−1^ were examined. The results suggest that ocean acidification suppressed the photosynthesis and increased the saturation irradiance for growth of the coccolithophore *Pleurochrysis carterae*. Further comparison with previously published results reveals that the physiological processes of the coastal coccolithophore specie *Pleurochrysis carterae* are less sensitive to ocean acidification than the smaller-sized species *Emiliania huxleyi* and *Gephyrocapsa oceanica*, and the saturation irradiance for the growth, particulate organic carbon (POC) and particulate inorganic carbon (PIC) production of *Pleurochrysis carterae* are much lower than those of the other two species.

**Abstract:**

The ecologically important marine phytoplankton group coccolithophores have a global distribution. The impacts of ocean acidification on the cosmopolitan species *Emiliania huxleyi* have received much attention and have been intensively studied. However, the species-specific responses of coccolithophores and how these responses will be regulated by other environmental drivers are still largely unknown. To examine the interactive effects of irradiance and ocean acidification on the physiology of the coastal coccolithophore species *Pleurochrysis carterae*, we carried out a semi-continuous incubation experiment under a range of irradiances (50, 200, 500, 800 μmol photons m^−2^ s^−1^) at two CO_2_ concentration conditions of 400 and 800 ppm. The results suggest that the saturation irradiance for the growth rate was higher at an elevated CO_2_ concentration. Ocean acidification weakened the particulate organic carbon (POC) production of *Pleurochrysis carterae* and the inhibition rate was decreased with increasing irradiance, indicating that ocean acidification may affect the tolerating capacity of photosynthesis to higher irradiance. Our results further provide new insight into the species-specific responses of coccolithophores to the projected ocean acidification under different irradiance scenarios in the changing marine environment.

## 1. Introduction

Since the beginning of the industrial revolution, human activities, such as the combustion of fossil fuels and destruction of forest vegetation, have caused a significant emission of CO_2_ into the atmosphere. In 2019, the global average of the atmospheric CO_2_ concentration reached 410 ppm [1]. Models have predicted that the atmospheric CO_2_ concentration will continue increase to 800–1000 ppm by the end of this century [2]. The increase in CO_2_ concentration has a crucial impact on the ecological environment, affecting the global climate and triggering the greenhouse effect. For marine ecosystems, the increase in CO_2_ concentration has consequences in the seawater carbonate system [3], lowering the pH levels of surface seawater, and leading to ocean acidification (OA). Warming may also cause intensified stratification in the oceans, resulting in a shallower mixed-layer depth and thus increasing the irradiance level for marine organisms living in the photic zone [4]. Daily irradiance has been reported to range from 77 to 740 μmol photons m^−2^ s^−1^ in shallow mixed coastal waters and is predicted to increase owing to climate change [5]. Increased CO_2_ concentration and irradiance bring significant challenges to marine ecosystems.

Marine coccolithophores are a type of calcifying unicellular marine phytoplankton, producing particulate inorganic carbon (PIC) through calcification and particulate organic carbon (POC) through photosynthesis [6]. It is estimated that coccolithophores contribute ~1–10% of the oceanic net primary production and 40–60% of the marine CaCO_3_ production [7,8,9], thus playing an important role in the marine carbon cycle. Previous studies suggest that the calcification of coccolithophores is especially susceptible to OA, but these responses vary between strain or species specificity [10,11]. In general, OA may be detrimental to the coccolith formation and maintenance [12], resulting in decreased calcification rates and the malformation of coccoliths [13,14,15,16,17]. However, the calcification of the *Emiliania huxleyi* strain NZEH increased significantly with CO_2_ concentration, increasing from 490 to 750 ppm [18], while the calcification rate of *Coccolithus pelagicus* remained constant across a range of CO_2_ concentrations [11]. This strain/species specific response may be related to the genetic diversity between different coccolithophore species/strains, due to their wide distribution in the world’s oceans [10]. Coccolithophores have been reported to exist in all marine ecosystems from equatorial to sub-polar regions, ranging from nutrient-rich upwelling zones to nutrient-poor zones, and from surface waters to depths greater than 100 m [19].

On the other hand, irradiance is another critical factor impacting coccolithophore physiology, providing the energy source for calcification and photosynthesis [20]. Coccolithophore blooms in the field are generally observed in environments with a high irradiance [21]. Previous studies have mainly focused on the sensitivity of the cosmopolitan species *Emiliania huxleyi* and *Gephyrocapsa oceanica* to environmental changes [13,22,23]. Nonetheless, marine coccolithophores have a global distribution, with a diversity of ~280 morphospecies [24]. As such, different coccolithophore species may respond to changes in irradiance and OA differently. For instance, a decrease in the cellular particulate organic carbon (POC) of *E. huxleyi* strain PML B92/11 was observed under the OA condition, regardless of the low (54 μmol photons m^−2^ s^−1^) or high (456 μmol photons m^−2^ s^−1^) irradiance the cells were exposed to; while the change in irradiance alone showed no significant effects on the POC content. However, the POC content of the coccolithophore *Scyphosphaera apsteinii* decreased when exposed to a low irradiance (50 μmol photons m^−2^ s^−1^) [19]. Therefore, it is important to study the effects of the change in CO_2_ concentration and irradiance on coccolithophore physiology and investigate the differences among different coccolithophore species/strains.

To investigate the species-specific responses of coccolithophores to the interplay of OA and irradiance, we examined the effects of increased CO_2_ concentration under a range of irradiances (50, 200, 500, 800 μmol photons m^−2^ s^−1^) on the physiology of a coastal coccolithophore species *Pleurochrysis carterae*, by conducting laboratory semi-continuous incubation. We then reanalyzed previously published results on the coccolithophores *Emiliania huxleyi* and *Gephyrocapsa oceanica* and further compared the effects of irradiance under present and future predicted CO_2_ conditions on the calcification and photosynthesis between these coccolithophore species.

## 2. Materials and Methods

### 2.1. Experimental Setup

The marine calcifying coccolithophore species *Pleurochrysis carterae* was isolated from the western Bohai Sea after gradient dilutions in 96-well microplates in the f/2 medium in 2011, which was maintained as a stock batch culture under controlled conditions of 15 °C and an irradiance of 120 μmol photons m^−2^ s^−1^. The light/dark cycle was 12L:12D. The culture medium was prepared by sterilized 0.2 μm filtered natural seawater, and enriched with nutrient stock solutions and vitamins to the f/20 level (10-times dilution of f/2) [25]. 

The *P. carterae* cultures were inoculated into 500 mL Nalgene polycarbonate bottles (3 replicate bottles for each treatment) under a light/dark cycle of 12/12 h with a range of irradiances (50, 200, 500 and 800 μmol photons m^−2^ s^−1^) at CO_2_ concentrations of 400 ppm (ambient CO_2_ treatment, LC) and 800 ppm (OA treatment, HC), respectively, in a thermo-controlling incubator (GXZ-1000C, Ningbo, China). The light source comprised light-emitting diodes (LED) lamps. The cell densities were monitored daily to indicate the growth. For the semi-continuous incubation, daily dilutions were performed using f/20 medium [17,22] to adjust the biomass to that of the previous day. After the cells were pre-acclimated at the experimental irradiance and CO_2_ concentration conditions for 3 days, the cell growth was maintained in the exponential phase throughout the incubation.

Irradiance was measured using a LI-1500 date logger (LI-COR, Biosciences, Lincoln, NE, USA). The f/20 medium was pre-aerated with a filtered air/CO_2_ and air mixture to achieve the CO_2_ concentrations in the ambient (400 ppm)/OA (800 ppm) treatments. During the manipulation experiments, the seawater carbonate chemistry was adjusted by the constant bubbling of an ambient air/CO_2_ and air mixture into each incubation bottle [26]. The CO_2_ and air mixture were obtained using a CO_2_-enriching device (CW100B, Ruihua, Wuhan, China). The incubation experiment was started with a low cell density of ~10^4^ cells mL^−1^ to maintain an optically thin cell density and minimize the photosynthetic effects on the carbonate chemistry in the medium and cell self-shadings. The seawater pH in each incubation bottle was measured daily within the first 2 h of the light period. The f/20 medium used for daily dilution in the semi-continuous incubation was also pre-aerated in order to maintain relatively constant *p*CO_2_ in each experimental treatment [27]. 

Final sampling was conducted when the steady growth phase was reached, with a variation in the growth rates of less than 10% for at least 7 generations, where the whole incubation period was ~21 days [28].

### 2.2. Sample Analysis

#### 2.2.1. Carbonate Chemistry Measurements

Seawater salinity was determined using an optical salinity meter (LS10T, Ruiming, Shanghai, China). The total dissolved inorganic carbon (DIC) was determined using a total organic carbon analyzer (TOC-LCPH, Shimadzu, Kyoto, Japan). The pH level was determined at 15 °C using a pH meter (SevenCOmpactTM S210K, Mettler Toledo, Greisensee, Switzerland), calibrated with NBS calibration solutions. Total alkalinity (TA) was determined by performing potentiometric titrations on filtrated samples (0.6 μm) [29]. The carbonate chemistry in the incubation system was estimated using the program CO2SYS based on TA, pH, temperature, salinity and phosphate concentration during the beginning, middle and final sampling of the incubation [29].

#### 2.2.2. Cell Density, Growth Rate and Chlorophyll a

For cell counts, 6 μL of modified Lugol’s solution was added into 1 mL sample [30] and the samples were stored at 4 °C. The cell density was then measured using a nanoplankton counting chamber under a microscope (CH20BIMF200, Olympus, Tokyo, Japan).

The growth rate (μ) was calculated according to the equation [31]:(1)$$µ=(\ln N_{n}-\ln N_{n-1})/(T_{n}-T_{n\text{-}1})$$
where N_n_ and N_n−1_ are the cell density at the beginning of the nth dilution (T_n_) and after the (n − 1)th dilution (T_n−1_).

Chl.-a was extracted using 90% acetone solution at ~20 °C for 12 h. The samples were vigorously shaken in the dark before being analyzed using the acidification method with a fluorometer (Trilogy, Turner Designs, San Jose, CA, USA) [32].

#### 2.2.3. Elemental Contents and Cell Size

Two sets of samples were filtered on pre-combusted GF/F filters (450 °C, 4 h), with one stored for total particulate carbon (TPC) and particulate organic nitrogen (PON) determination, and another for particulate organic carbon (POC) determination. The sample for POC determination was fumed with concentrated HCl for 3 h to remove inorganic carbon before analysis. The samples for TPC, PON and POC were measured using a CHN element analyzer (ECS4010, Costech, Milan, Italy). The particulate inorganic carbon (PIC) was the difference between the TPC and POC [33]. The particulate organic phosphorus content (POP) was determined by the phosphorus molybdenum blue spectrophotometric method [34]. The POC production rate (POC Prod) and PIC production rate (PIC Prod) were calculated as:POC Prod = μ × POC cell^−1^(2)
PIC Prod = μ × PIC cell^−1^(3)

The cell size (size of the whole coccosphere) was measured using a laser granulometer (LS 13320, Beckman, Brea, CA, USA). The detailed values of the test are presented in Appendix A.

#### 2.2.4. Data Analysis and Fitting

The individual or interactive effects of CO_2_ concentration and irradiance on all physiological parameters were analyzed with a two-way analysis of variance (ANOVA) using GraphPad Prism 8.0 software. The significance between different treatments was tested by Tukey’s post-hoc multiple comparisons test. The effects of CO_2_ concentration (LC and HC) on the saturated irradiance and the maximum values of growth, POC and PIC production rates were compared using a *t*-test. All significance levels were evaluated at the *p* < 0.05 level. 

The growth rate (μ) vs. irradiance curves were fitted using a model of Eilers and Peters [35]:(4)$$µ=1/(a\;I^{2}+b\;I+c)$$
(5)$${µ}_{\max}=1/(b+2\sqrt{\mathrm{ac}})$$
(6)α=1/c
(7)$$I_{k}=\sqrt{a/c}$$
where I represents the irradiance level; µ_max_ is the growth rate to the saturated irradiance (I_k_); a, b and c are the irradiance response constants; and α represents the light-use efficiency. 

The POC and PIC production rate vs. irradiance curves (P-I curves) were fitted to the Steele model [36]:(8)$$P=\alpha {\;P}_{\max}\;I\;\exp(1\;\text{-}\;\alpha I)$$
(9)$$I_{k}=I/\alpha$$
where $P_{\max}$ is the photosynthetic and calcification rate to the saturated irradiance (I_k_).

The OA-induced inhibition was calculated as:(10)$$\mathrm{Inhibition}\;\mathrm{rate}(\%)=(Y_{HC}-Y_{LC})/Y_{LC}\times 100$$
where Y is the value of growth or POC or PIC production rate at different CO_2_ concentration treatments.

## 3. Results

### 3.1. Carbonate System in Experiments

The carbonate system on the final sampling day in the media is presented in Table 1. There was a significant difference in the carbonate chemistry of pH values of 8.16 ± 0.01 (LC) and 7.93 ± 0.02 (HC), with corresponding CO_2_ concentrations of 420 ± 9 and 797 + 22 ppm (*p* < 0.05).

### 3.2. Growth, POC and PIC Production Rate

A one-way ANOVA analysis indicated that irradiance significantly affected the growth, POC and PIC production rates of *P. carterae* (*p* < 0.05), which increased with increasing irradiance and then declined (Figure 1). Photo-inhibition was observed at a high irradiance (HL = 800 μmol photons m^−2^ s^−1^) for growth, POC and PIC production rates, with reductions by 28.07%, 75.07% and 76.01% compared to the medium irradiance treatment (ML = 200 μmol photons m^−2^ s^−1^), respectively (*p* < 0.05).

OA weakened the POC production rate at a low irradiance (LL = 50 μmol photons m^−2^ s^−1^, Figure 1B,E, *p* < 0.05). The negative effect of OA on POC production rate was decreased with increased irradiance, and the percentages of inhibition were 44.12%, 19.20%, 15.91% and 5.88%, respectively (Figure 1E, *p* < 0.05). In addition, OA also slightly weakened the PIC production (Figure 1C,F, *p* > 0.05).

### 3.3. Saturated Irradiance and the Maximum Value of Growth, POC and PIC Production Rates

OA increased the calculated saturating irradiance (I_k_) for growth, POC and PIC production rates, among which, the change in I_k_ for the growth rate was the largest (Figure 2A, *t*-test, *p* < 0.05), with an increase from 103 ± 13 μmol photons m^−2^ s^−1^ to 158 ± 7 μmol photons m^−2^ s^−1^. 

On the contrary, OA reduced the maximum POC production rate (POC Prod_max_) in *P. carterae* (from 5.10 ± 0.24 to 3.84 ± 0.19 pg cell^−1^ d^−1^, Figure 2B, *p* < 0.05). 

### 3.4. The Chl.-a Content

The relationship between cellular Chl.-a content and irradiance was similar to that of growth rate (Figure 3). The cellular Chl.-a content was highest at 200 μmol photons m^−2^ s^−1^ and then decreased significantly at 500 and 800 μmol photons m^−2^ s^−1^ (*p* < 0.05).

### 3.5. Elemental Composition

The POC, PIC, PON and POP contents of *P. carterae* were significantly affected by irradiance, with the highest values at 200 μmol photons m^−2^ s^−1^ and then declined sharply in the experimental irradiance range (Figure 4, Table 2). It was noteworthy that a high irradiance reduced the cellular elemental contents compared to ML treatment (*p* < 0.05). On the contrary, the PIC:POC ratio increased with increasing irradiance and peaked at 500 μmol photons m^−2^ s^−1^, which was about three-times higher than those of *P. carterae* at 50 μmol photons m^−2^ s^−1^ (Figure 4H).

The results showed that there was significant effect of OA on the contents of POC and PON at a low irradiance (50 μmol photons m^−2^ s^−1^, Figure 4A,C, *p* < 0.05). Cellular POC and PON decreased by 42.00% and 28.84% (*p* < 0.05), respectively, in HC compared with LC treatments.

## 4. Discussion

This study revealed that a high irradiance had negative effects on the physiology of *P. carterae*. In addition, OA resulted in an elevation of the saturation irradiance for the growth rate, while causing a reduction in the rate of POC production under conditions of low irradiance. This inhibitory effect of OA was found to diminish as irradiance levels increased. 

### 4.1. High Irradiance Inhibited the Physiological Processes

Irradiance is considered to be an important physical factor controlling the metabolic activity and growth of algae [22]. In the present study, both the POC and PIC production rates of *P. carterae* were positively correlated with the available energy under unsaturated irradiance but became negatively correlated and decreased when the irradiance exceeded the metabolic capacity. This trend was consistent with previous findings [37]. When irradiance is higher than saturation, the negative effects of a. high irradiance cause damage to Photosystem I (PSI) and Photosystem II (PSII) and PSII is more sensitive to environmental changes [38]. The photosynthetic system is damaged by the reactive oxygen species (ROS) produced by the antenna complex when it enters the triplet state during light absorption [39,40], and large amounts of ROS cause oxidative damage to the D1 protein of the PSII reaction center at a rate that exceeds the repair rate, resulting in a significant reduction in PSII activity and photo-inhibition (Figure 5) [41]. On the other hand, under conditions of photo-inhibition, the non-photochemical quenching (NPQ) process is significantly upregulated [42], leading to an increase in energy allocation towards photo-protective mechanisms aimed at dissipating excess excitation energy [43].

### 4.2. Acidification Affected the POC, PIC Production and the Elemental Compositions 

Our results suggest that OA weakened the photosynthesis of *P. carterae*. This negative effect is primarily due to OA reducing the activity of H^+^ channels [44], resulting in the accumulation of high concentrations of H^+^ within the cell. The cells require more energy to maintain homeostasis by removing the excess H^+^ that accumulates in the cytoplasm to prevent acidosis and cell death [45]. OA leads to an increase in the mitochondria respiration rate of the coccolithophore [46]. Studies have found that the *E. huxleyi* (strain CCMP 1516) mitochondria respiration rate increased by about 130% under OA conditions (1000 uatm) [46]. In addition, the increase in H^+^ concentration in the chloroplast stroma may also lead to a decrease in the CO_2_ fixation efficiency [47,48]. Due to the increased energy consumption under OA treatment, the algae increase the demands for energy, resulting in a decrease in the POC production rate. 

The increase in intracellular H^+^ concentration also slightly decreased calcification and the cell size of *P. carterae* (Appendix A). According to physiological evidence, HCO_3_^−^ is the primary substrate for calcification [49,50] and the nucleation of calcium carbonate occurs by transferring the substrate (dissolved inorganic carbon and Ca^2+^) to the coccolith vesicle (CV) while removing soluble products, especially H^+^ [51]. To maintain an environment favorable for the precipitation of calcite, H^+^ must be released from CV into the cytosol [52]. OA reduces H^+^ channel activity, resulting in an increase in H^+^ concentration in the cytosol, which decreases the transmembrane H^+^ electrochemical gradient and lowers the rate of H^+^ efflux [53]. This increases the cost of calcification, ultimately inhibiting calcification. Although our results showed that OA inhibited the calcification of *P. carterae*, the effect was not statistically significant. It is noteworthy that the *P. carterae* used in our study was isolated from a coastal environment, with seawater chemistry fluctuating more than that in the oceanic environment. This likely resulted in a higher adaptive capacity of *P. carterae* to OA than *E. huxleyi*. A previous study has suggested that the calcification process of *E. huxleyi* could be sensitive to OA, with malformed coccoliths observed [54]. 

In addition, the cellular contents of POC and PON of *P. carterae* were both decreased under OA. A rising CO_2_ concentration may affect gene expression, enzyme kinetics and metabolic fluxes in the cells, resulting in changes in the relative requirements of these elements in the phytoplankton [20,55].

While OA reduced the carbon-fixation rate of *P. carterae*, the growth was slightly promoted under the OA condition. This implied that the coccolithophore *P. carterae* may respond to OA by modifying its substrate uptake rates and increasing cell division. This finding is consistent with a previous study showing that an increase in *p*CO_2_ from 40 Pa (~395 ppm) to 80 Pa (~790 ppm) reduced the photosynthetic and calcification rates but promoted the growth of *E. huxleyi* [22].

### 4.3. Interactive Effects of OA and Irradiance

The present study shows that irradiance regulated the responses of *P. carterae* to acidification. Under OA, the saturated irradiance for growth increased compared to the ambient CO_2_ condition, indicating that the tolerating capacity to higher irradiance of *P. carterae* will be strengthened under future OA scenarios. In addition, OA had different effects under different irradiances. Specifically, OA negatively affected the POC production rate of *P. carterae* at a low irradiance and the negative effect decreased with increasing irradiance. In contrast, a previous study revealed that the POC quota of the coccolithophore *E. huxleyi* was not significantly affected by OA across a range of irradiances (80~200 μmol photons m^−2^ s^−1^) in the nutrient-rich environment [56], suggesting species-specific responses [11].

**Figure 5 biology-12-01249-f005:**
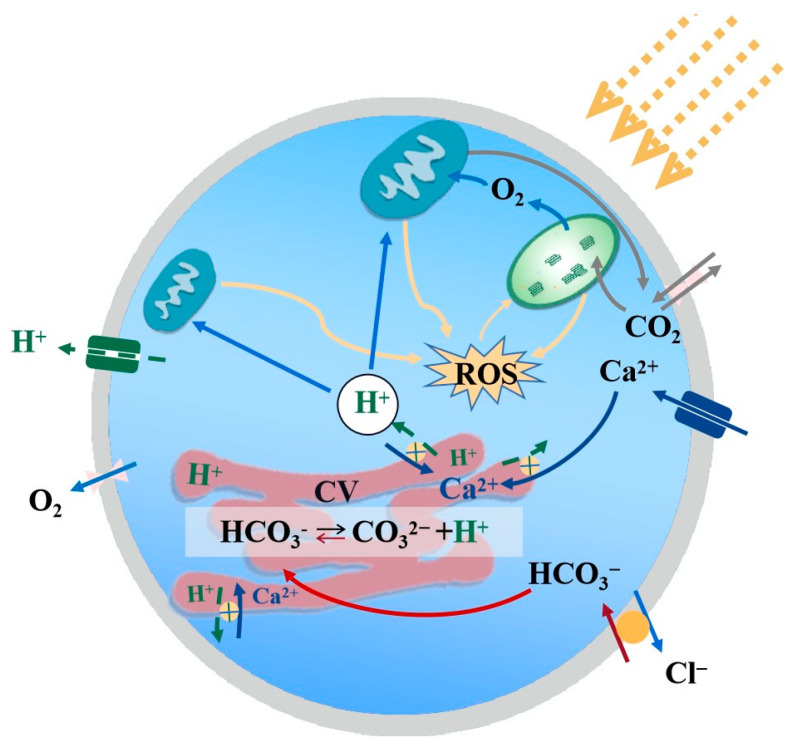
The mechanism of OA and high irradiance acting on the physiological activities of coccolithophores. The graph is modified from studies of calcification and photosynthetic pathway analysis [45,57], with green lines indicating that the process is inhibited by OA and yellow indicating that the process is affected by a high irradiance.

### 4.4. Species-Specific Responses of Coccolithophores

By fitting previously published results of OA and irradiance effects on other coccolithophore species, interspecies responses were observed. The saturating irradiance for the POC and PIC production rates of both *E. huxleyi* (calculated based on data reported by Jin et al., 2017 [58]) and *G. oceanica* (calculated based on data in Zhang et al., 2015 [59]) were higher than those of the larger-celled species *P. carterae* (Figure 6). Furthermore, the saturated irradiance and the corresponding growth and POC production rates of *E. huxleyi* increased under OA, while the response of *P. carterae* was rather smaller. Contrarily, those of *G. oceanica* decreased with the increased CO_2_ concentration. 

It is speculated that the differential responses to acidification among different coccolithophore species and strains are likely due to the fundamental differences in calcification pathways, components and energy allocation [51,57]. From a genetic perspective, this may be caused by the genotypic variations among different coccolithophore species [51], which are associated with the different environmental conditions at the initial isolation sites [60]. However, the current available genetic information is still limited, and further investigation is still needed to explore the underlying mechanisms [61,62]. In addition, the study on *E. huxleyi* selected for the comparison was based on outdoor incubation under natural solar radiation [58], while our study on *P. carterae* and that of Zhang et al. on *G. oceanica* [59] were both conducted in the laboratory. The difference in light source and irradiance levels may also have caused the differential responses.

## 5. Conclusions

In summary, our results indicate that OA affects the response of the coastal coccolithophore species *P. carterae* to irradiance. An increase in CO_2_ concentration weakened the POC productivity of *P. carterae*, especially under a low irradiance, and increased the saturation irradiance of growth. Comparisons with previously published results reveal that the physiological processes of the coastal coccolithophore species *P. carterae* are less sensitive to the increase in CO_2_ concentration and irradiance than the smaller-sized species *E. huxleyi* and *G. oceanica*, which may lead to changes in the ecological niche and community structure of natural coccolithophores in the future. Our results not only highlight the necessity to consider other environmental conditions, such as irradiance, when examining the OA effects on the physiology of marine coccolithophores, but also provide insight into the species-specific response patterns of coccolithophores to complex environmental changes.

## Figures and Tables

**Figure 1 biology-12-01249-f001:**
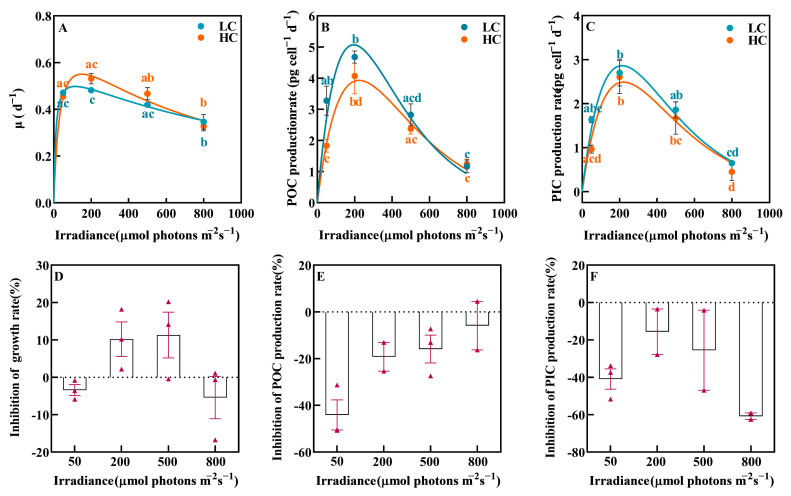
The growth rate (**A**), POC production (**B**) and PIC production (**C**) rates of *P. carterae* under different CO_2_ and irradiance treatments. The inhibition of OA on growth (**D**), POC production (**E**) and PIC production (**F**) rates. Values shown are the mean ± SEM of triplicate samples. Different letters represent significant differences between different irradiance or CO_2_ treatments (*p* < 0.05). The triangles represent specific values.

**Figure 2 biology-12-01249-f002:**
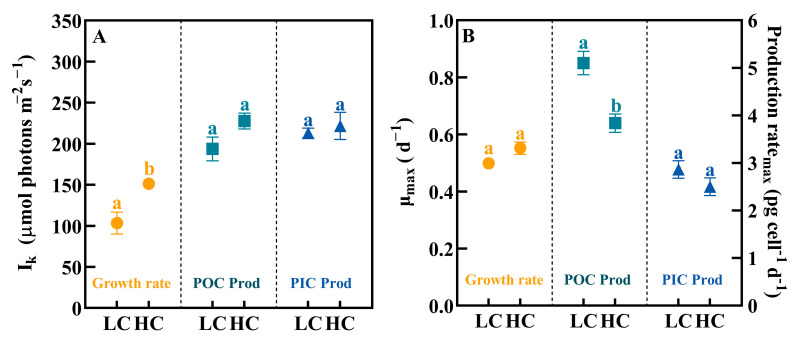
The saturating irradiance (I_k_, (**A**)) and the maximum values (**B**) of growth, POC and PIC production rates of *P. carterae* under different CO_2_ and irradiance treatments. Values shown are the mean ± SEM of triplicate samples. Different letters represent significant differences between different CO_2_ treatments (*p* < 0.05).

**Figure 3 biology-12-01249-f003:**
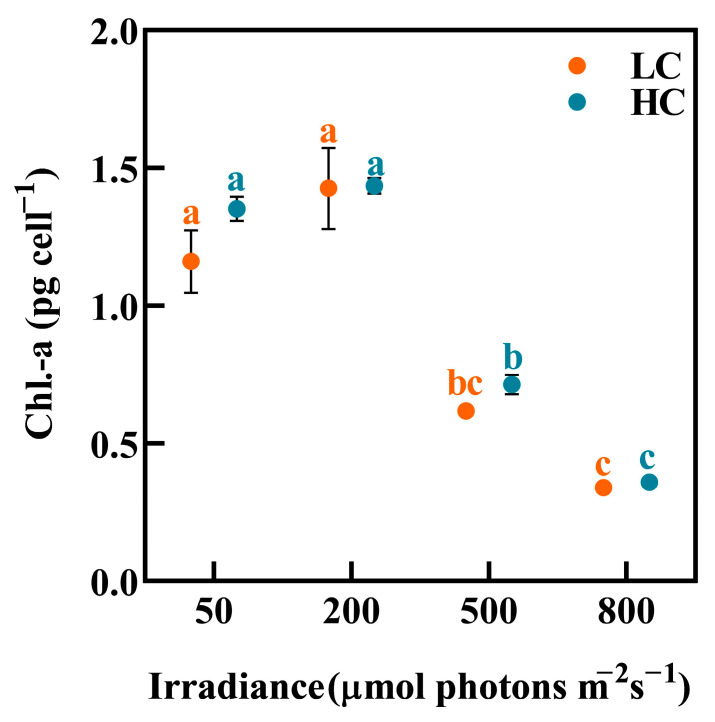
The content of Chl.-a of *P. carterae* under different CO_2_ and irradiance treatments. Values shown are the mean ± SEM of triplicate samples. Different letters represent significant differences between different irradiance or CO_2_ treatments (*p* < 0.05).

**Figure 4 biology-12-01249-f004:**
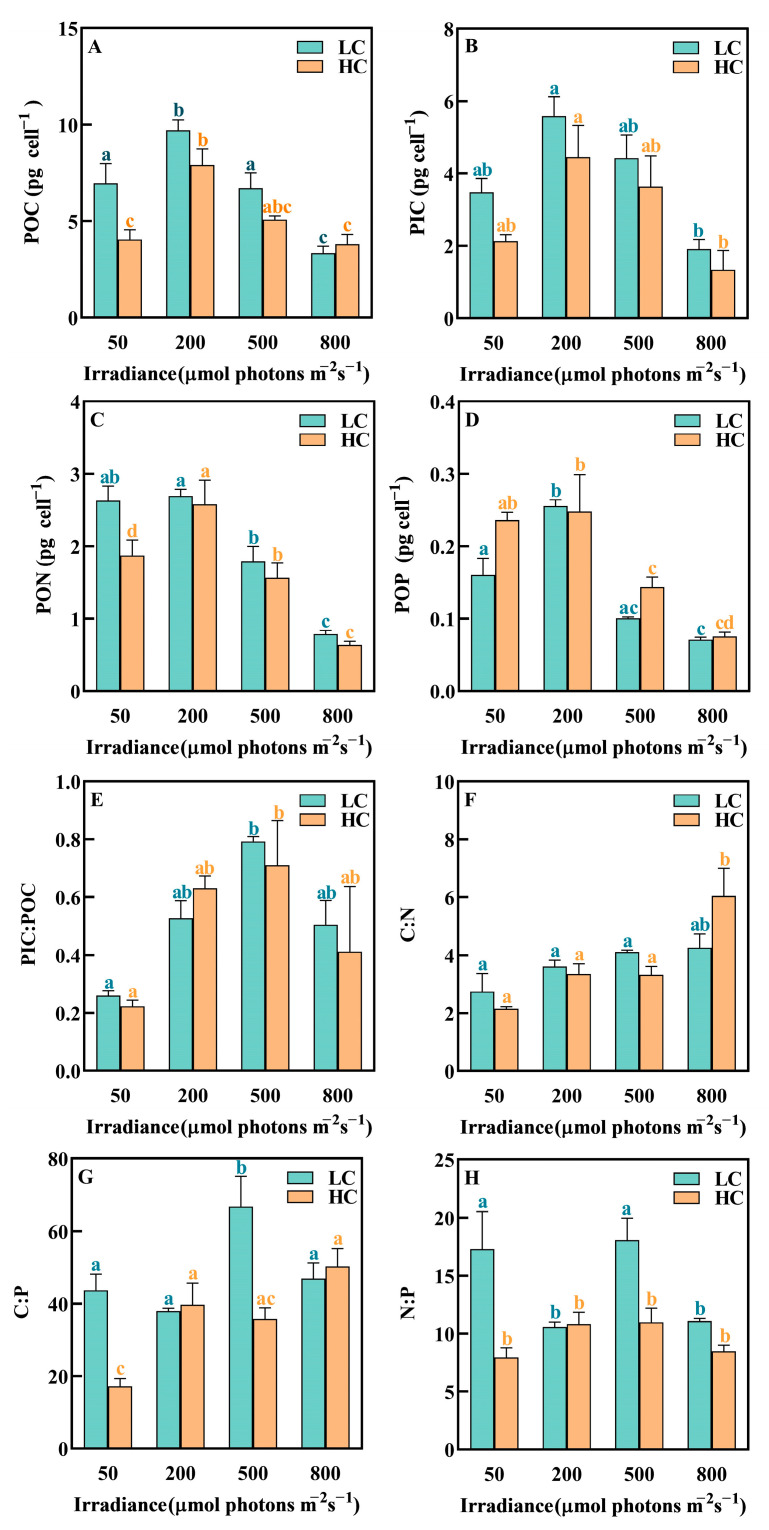
The elemental composition of *P. carterae* under different CO_2_ and irradiance treatments. (**A**) POC content; (**B**) PIC content; (**C**) PON content; (**D**) POP content; (**E**) PIC-to-POC ratio (PIC:POC); (**F**) POC-to-PON ratio (C:N); (**G**) POC-to-POP ratio (C:P); (**H**) PON-to-POP ratio (N:P). Values shown are the mean ± SEM of triplicate samples. Different letters represent significant differences between different irradiance or CO_2_ treatments (*p* < 0.05).

**Figure 6 biology-12-01249-f006:**
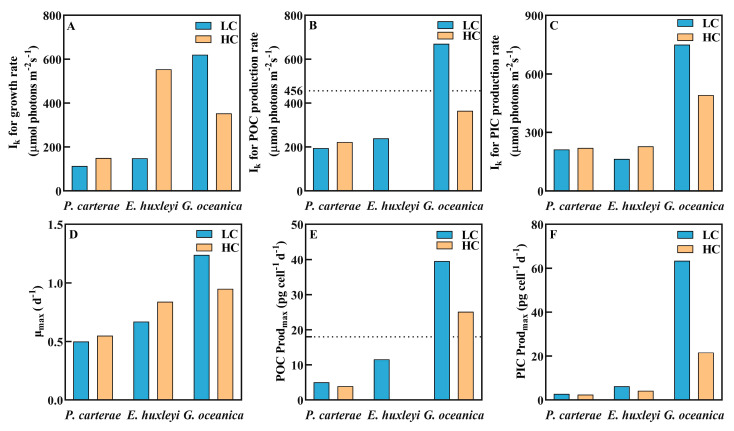
Parameters of irradiance-response curves of coccolithophores *P. carterae*, *E. huxleyi* and *G. oceanica*. The calculation for *E. huxleyi* was based on published data in Jin et al. [58] (condition of 20 °C, 398 and 1000 ppm CO_2_ selected). The calculation for *G. oceanica* was based on data extracted from Zhang et al. [59] (*p*CO_2_ 51 Pa (~500 ppm) and 105 Pa (~1000 ppm)). All data presented in the figures are the mean values of replicate samples (Jin et al., n = 3; Zhang et al., n = 4). (**A**) The saturation irradiance for the growth rate (μ); (**B**) the saturation irradiance for the POC production rate; (**C**) the saturation irradiance for the PIC production rate; (**D**) the maximum value of the growth rate (μ_max_); (**E**) the maximum value of the POC production rate (POC Prod_max_); (**F**) the maximum value of the PIC production rate (PIC Prod_max_). The growth rate vs. irradiance curves were fitted using a model of Eilers and Peters (Equation (4)). The POC and PIC production rate vs. irradiance curves (P-I curves) were fitted to the Steele model (Equation (8)), while the POC production rate vs. irradiance curve of *E. huxleyi* was fitted to the model of Eilers and Peters (Equation (4)). Based on the data fitting, no saturation point of POC production occurred within the experimental range of irradiances (54~456 μmol photons m^−2^ s^−1^) for *E. huxleyi* at the OA condition. Detailed information is presented in Appendix A.

**Table 1 biology-12-01249-t001:** The seawater carbonate chemistry on the final sampling day. pH and DIC (total inorganic carbon) were directly measured values. TA, [HCO_3_^−^], [CO_3_^2−^] and CO_2_ were calculated using CO2SYS. Values shown are the mean ± SEM of triplicate samples. Different letters represent significant differences between different irradiance or CO_2_ treatments (*p* < 0.05).

Treatment	pH_NBS_	TA μmol kg^−1^	DIC μmol kg^−1^	[HCO_3_^−^] μmol kg^−1^	[CO_3_^2−^] μmol kg^−1^	CO_2_ ppm
50 μmol photons m^−2^·s^−1^ +400 ppm	8.16 ± 0.03 ^a^	2303 ± 8 ^a^	2083 ± 5 ^a^	1927 ± 11 ^a^	140 ± 8 ^a^	420 ± 29 ^a^
50 μmol photons m^−2^·s^−1^ +800 ppm	7.91 ± 0.01 ^b^	2354 ± 3 ^b^	2229 ± 2 ^b^	2111 ± 2 ^b^	86 ± 2 ^bc^	812 ± 17 ^b^
200 μmol photons m^−2^·s^−1^ +400 ppm	8.17 ± 0.00 ^a^	2299 ± 12 ^a^	2074 ± 9 ^a^	1916 ± 8 ^a^	143 ± 2 ^a^	405 ± 3 ^a^
200 μmol photons m^−2^·s^−1^ +800 ppm	7.98 ± 0.02 ^c^	2265 ± 8 ^a^	2135 ± 16 ^c^	2045 ± 8 ^cd^	77 ± 0 ^b^	857 ± 8 ^b^
500 μmol photons m^−2^·s^−1^ +400 ppm	8.16 ± 0.02 ^a^	2278 ± 6 ^ab^	2062 ± 6 ^ab^	1909 ± 11 ^a^	137 ± 7 ^a^	420 ± 27 ^a^
500 μmol photons m^−2^·s^−1^ +800 ppm	7.88 ± 0.01 ^b^	2277 ± 39 ^a^	2159 ± 11 ^cd^	2015 ± 21 ^c^	94 ± 8 ^c^	685 ± 39 ^c^
800 μmol photons m^−2^·s^−1^ +400 ppm	8.14 ± 0.01 ^a^	2239 ± 15 ^b^	2035 ± 13 ^b^	1889 ± 11 ^a^	129 ± 2 ^a^	435 ± 6 ^a^
800 μmol photons m^−2^·s^−1^ +800 ppm	7.91 ± 0.01 ^bc^	2296 ± 7 ^a^	2173 ± 3 ^d^	2059 ± 2 ^d^	84 ± 2 ^bc^	798 ± 17 ^b^

**Table 2 biology-12-01249-t002:** Results of two-way ANOVA of the effects of irradiance and CO_2_ and their interaction on growth rate, Chl.-a and POC, PIC, PON and POP contents, PIC/POC, C/N, C/P and N/P ratios, POC production and PIC production. Asterisk indicates significance at the *p* < 0.05 level.

Parameters	CO_2_			Irradiance			Irradiance × CO_2_		
	df	F	*p*	df	F	*p*	df	F	*p*
Growth rate	1	1.30	0.27	3	28.27	<0.01 *	3	1.97	0.16
Chl.-a	1	2.51	0.13	3	103.2	<0.01 *	3	0.72	0.55
POC	1	10.69	0.01 *	3	21.55	<0.01 *	3	2.61	0.09
PIC	1	4.94	0.04 *	3	12.69	<0.01 *	3	0.16	0.92
PON	1	5.22	0.04 *	3	39.43	<0.01 *	3	1.27	0.32
POP	1	3.82	0.07	3	28.75	<0.01 *	3	1.65	0.22
C/N	1	0.01	0.93	3	10.23	<0.01 *	3	2.78	0.08
C/P	1	15.22	<0.01 *	3	8.51	<0.01 *	3	7.38	<0.01 *
N/P	1	20.22	<0.01 *	3	3.85	0.03 *	3	4.49	0.02 *
PIC/POC	1	0.05	0.83	3	1.88	0.17	3	0.20	0.89
POC Prod	1	8.26	0.01 *	3	35.32	<0.01 *	3	2.35	0.11
PIC Prod	1	2.91	0.11	3	27.32	<0.01 *	3	0.59	0.63

## Data Availability

Not applicable.

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
