# Peer review of "Ocean Acidification Affects the Response of the Coastal Coccolithophore Pleurochrysis carterae to Irradiance"

_biology, 2023, doi:10.3390/biology12091249_

Round 1
Reviewer 1 Report (Previous Reviewer 2)
The authors improved the overall quality of the manuscript and were not limited to just the comments highlighted by the reviewers. Only a quick grammar check of the manuscript is necessary (but this can be done while proofreading).
Minor editing is required
Reviewer 2 Report (Previous Reviewer 4)
The authors responded adequately.
This manuscript is a resubmission of an earlier submission. The following is a list of the peer review reports and author responses from that submission.
Round 1
Reviewer 1 Report
The article Ocean acidification affected the response of coastal coccolithophore Pleurochrysis carterae to irradiance deals with experimentation on this phytoplankton member under ocean acidification and different irradiation conditions to establish the different responses on coccolithophore species.
The approach is interesting, the methods seem to be valid, though more information is needed, and the conclusions seem to be supported by the results. This article should be published; however, there are some comments that might have to be addressed.
Graphical Abstract:
It is not clear what the authors mean by “semi-continuous” incubation. The authors must check the spelling in the image and explain here what LC and HC mean. The graphical abstract should be self-explanatory.
Throughout the manuscript: the authors should add a space between the words and the reference brackets, also between the number and °C.
Introduction
L65: please explain what NZEH means. It must be clear if this is the name of a specific strain.
L79: same comment; what does PML B92/11 means? It is very confusing.
Material and Methods
L96: Please add information on the isolation, such as coordinates, how it was sampled, and how the culture was established.
L99: why use f/20 medium? Please explain why that diluted.
L105: irradiance was measured, or irradiances were measured.
L110: please explain why this initial abundance was selected and how it was determined (how cells were counted).
L111: please explain how the cells were preadapted.
L114-115: please report how the cells were counted and how the abundance was determined. Also, please explain how the steady growth phase was determined.
L130: please explain what you mean by nanoplankton counting chamber.
L132: please explain how these time points were selected. What is T2 and T1? Were these random?
Results
The authors report maximum values, but according to their figures, most of them were not statistically significant. If this is the case, there were no differences (if the differences are not statistically significant, then there is no difference).
L174: content. A t is missing.
Figure 1. Please use the irradiance numbers that were used in the experiment. No significant differences exist among 0-600 umol phot m-2s-1, so the authors cannot claim or conclude on these supposed differences. Also, there are no differences in Chl a between 50 and 200 umol phot m-2s-1.
The authors must carefully check their results and rewrite them and their conclusions considering statistical significance.
Figure 4. Do not repeat ppm (line 236).
Discussion
Were the experiments performed on E. huxleyi and G. oceanica identical to these so that their results can be compared? Please discuss.
L290: please check your wording. High irradiance does not inhabit anything. I guess the authors meant inhibited.
Figure 5. Please provide a figure with a better resolution.
English language is fine and requires minor revisions, but there is an important lack of spaces before the references brackets and some punctuation mistakes.
Response to Reviewer 1 Comments
The article Ocean acidification affected the response of coastal coccolithophore Pleurochrysis carterae to irradiance deals with experimentation on this phytoplankton member under ocean acidification and different irradiation conditions to establish the different responses on coccolithophore species.
The approach is interesting, the methods seem to be valid, though more information is needed, and the conclusions seem to be supported by the results. This article should be published; however, there are some comments that might have to be addressed.
Response: Thank you very much for reviewing our manuscript and giving valuable comments. These comments are very helpful for improving our manuscript. According to your comments, we have thoroughly revised the manuscript. The responses to the specific comments are listed below.
Point 1: Graphical Abstract: It is not clear what the authors mean by “semi-continuous” incubation. The authors must check the spelling in the image and explain here what LC and HC mean. The graphical abstract should be self-explanatory.
Response 1: We have changed the “semi-continuous incubation” into “incubation system” and added a note of “LC and HC mean low and high CO2 treatment, respectively. “to explaining the “LC and HC” in graphical abstract.
Point 2: Throughout the manuscript: the authors should add a space between the words and the reference brackets, also between the number and °C.
Response 2: Thanks. Based on your comments, we have carefully checked the reference brackets and the °C symbols throughout the m/s and spaces have been added.
Point 3: L65: please explain what NZEH means. It must be clear if this is the name of a specific strain. L79: same comment; what does PML B92/11 means? It is very confusing.
Response 3: The NZEH and PML B92/11 are strain names for Emiliania huxleyi, respectively. The previous text has been modified for clarification in lines 68 and 84 in the submitted pdf file: “ However, the calcification of Emiliania huxleyi strain NZEH increased significantly……“ ” For instance, decrease in POC content of E. huxleyi strain PML B92/11 was observed under OA condition, ……“
Point 4: L96: Please add information on the isolation, such as coordinates, how it was sampled, and how the culture was established.
Response 4: Corrections have been made in lines 102-105: “The marine calcifying coccolithophore species Pleurochrysis carterae was isolated from the western Bohai Sea by gradient dilutions in 96-well microplates in the f/2 medium in 2011, which was maintained as a stock batch culture under controlled conditions of 15 °C and an irradiance of 120 μmol photons m−2 s−1. “
Point 5: L99: Why use f/20 medium? Please explain why that diluted.
Response 5:The bloom of coccolithophore was preceded after diatoms, which had low nutrient requirements, so the f/20 medium was used. This is also the commonly used medium for laboratory incubation of coccolithophores, therefore we have cited the following reference:
Feng, Y.; Roleda, M.Y.; Armstrong, E.; Summerfield, T.C.; Law, C.S.; Hurd, C.L.; Boyd, P.W. Effects of multiple drivers of ocean global change on the physiology and functional gene expression of the coccolithophore Emiliania huxleyi. Global Change Biology 2020, 26, 5630-5645.
Feng, Y.; Roleda, M.Y.; Armstrong, E.; Boyd, P.W.; Hurd, C.L. Environmental controls on the growth, photosynthetic and calcification rates of a Southern Hemisphere strain of the coccolithophore Emiliania huxleyi. Limnology and Oceanography 2017, 62, 519-540.
Point 6: L105: Iirradiance was measured, or irradiances were measured.
Response 6: Corrections have been made in line 118: “Irradiance was measured using a light sensor logger (LI-1500, LI-COR, USA).”.
Point 7: Please explain why this initial abundance was selected and how it was determined (how cells were counted).
Response 7: With high cell density, CO2 uptake by photosynthesis will severely affect the carbonate chemistry; on the other hand, self-shading of cells tends to decrease the actual irradiance that cells are exposed to. Therefore, relatively low initial cell abundance was selected to minimize these impacts. We have added the detailed information in the revised m/s as: “The incubation experiment was stared with low cell density of ~104 cells mL-1 to maintain an optically thin cell density and minimize the photosynthetic effects on the carbonate chemistry in the medium and cell self-shadings” in L124-L126.
The cells were counted used a nanoplankton counting chamber under microscope. The cell density was calculated using the formula y = x * k / v, where x represents the mean number of cells counted in a single field of view, k is the ratio of the counting frame area to the field of view area, and v is the volume of the counting frame. This section can be found in the article from L138 to L139:” The cell density was then measured using a nanoplankton counting chamber under a microscope (CH20BIMF200, Olympus, Japan).”
Point 8: L111: please explain how the cells were preadapted.
Response 8: The explanation has been added in L113-L116: “For the semi-continuous incubation, daily dilutions were performed using f/20 medium to adjust the biomass to that of the previous day, after the cells were pre-acclimated at the experimental irradiance and CO2 concentration conditions for 3 days.”.
Point 9: L114-115: please report how the cells were counted and how the abundance was determined. Also, please explain how the steady growth phase was determined.
Response 9: Experimental cell density was obtained by microscopic counting, which is the same method as in response 7. When the growth rate at a rate of less than 10% for more than 7 generations, it is in steady growth phase, the growth rate is calculated in lines 148 to 150. “The growth rate (μ) was calculated according to the equation [32]:
, where Nn and Nn-1 are cell density at the beginning of the nth dilution (Tn) and after the (n-1)th dilution (Tn-1).”
Reference:
Brading, P.; Warner, M.E.; Davey, P.; Smith, D.J.; Achterberg, E.P.; Suggett, D.J. Differential effects of ocean acidification on growth and photosynthesis among phylotypes of Symbiodinium (Dinophyceae). Limnology and Oceanography 2011, 56, 927-938.
Point 10: L130: please explain what you mean by nanoplankton counting chamber.
Response 10:The nanoplankton counting chamber is used to identify and count phytoplankton (algae) and small zooplankton in water samples under a light microscope with a volume of 0.1mL. As shown in the figure below.
Point 11: L132: please explain how these time points were selected. What is T2 and T1? Were these random?
Response 11:During the experiment, the samples were diluted daily. Due to the previous statement was unclear, and the changes were made in lines 148-150.
Point 12: The authors report maximum values, but according to their figures, most of them were not statistically significant. If this is the case, there were no differences (if the differences are not statistically significant, then there is no difference).
Response 12: We have recalculated the statistics and statistical significances are reported in the results section now. “Photo-inhibition was observed at high irradiance (HL = 800 μmol photons m-2 s-1) for growth, POC and PIC production rates, with reduction by 28.07%, 75.07% and 76.01% compared to medium irradiance treatment (ML = 200 μmol photons m-2 s-1), respectively (Figure 1, p<0.05).”” OA increased the calculated saturating irradiance(Ik) of growth, POC and PIC production rates, among which, the change in Ik for growth rate was the largest (Figure 2A, t-test, p < 0.05), with an increase from 103 ± 13 μmol photons m-2 s-1 to 158 ± 7 μmol photons m-2 s-1.”
Point 13: L174: content. A t is missing.
Response 13: Corrections have been made in manuscript.
Point 14: Figure 1. Please use the irradiance numbers that were used in the experiment. No significant differences exist among 0-600 μmol photons m−2 s−1, so the authors cannot claim or conclude on these supposed differences. Also, there are no differences in Chl a between 50 and 200 μmol photons m−2 s−1.
The authors must carefully check their results and rewrite them and their conclusions considering statistical significance.
Response 14: The results were recalculated and restated in the manuscript. “The cellular Chl.-a content was highest at 200 μmol photons m-2 s-1 and then decreased significantly at 500 and 800 μmol photons m-2 s-1(p < 0.05).”
Point 15: Figure 4. Do not repeat ppm (line 236).
Response 15: Corrections have been made in lines 360.
Point 16: Were the experiments performed on E. huxleyi and G. oceanica identical to these so that their results can be compared? Please discuss.
Response 16: Differences in experimental methodology may have an impact on the results of the experiment, and a discussion of the differences in results caused by differences in experimental methodology has been added in lines 339-343.” In addition, the study on E. huxleyi selected for the comparison was based on outdoor incubation under natural solar radiation [59], while our study on P. carterae and Zhang et al. on G. oceanica [60] were both conducted in the laboratory. The difference in light source and irradiance levels may have also caused the differential responses.”
References
Jin, P.; Ding, J.; Xing, T.; Riebesell, U.; Gao, K. High levels of solar radiation offset impacts of ocean acidification on calcifying and non-calcifying strains of Emiliania huxleyi. Marine Ecology Progress 2017, 568, 47-58.
Zhang, Y.; Bach, L.T.; Schulz, K.G.; Riebesell, U. The modulating effect of light intensity on the response of the coccolithophore Gephyrocapsa oceanica to ocean acidification. Limnology and Oceanography 2015, 60, 2145-2157.
Point 17: L290: please check your wording. High irradiance does not inhabit anything. I guess the authors meant inhibited.
Response 17: We have corrected the mis-spelling and moved this sentence to discussion section 4.1.
Point 18: Figure 5. Please provide a figure with a better resolution.
Response 18: The resolution of the figure has been improved.
Reviewer 2 Report
I have reviewed the manuscript entitled "Ocean acidification affected the response of coastal coccolithophore Pleurochrysis carterae to irradiance" submitted to Biology. The manuscript addresses the synergistic approach between the effects of increased pCO2 and irradiance on the growth and photosynthetic parameters of P. carterae. Additionally, the results were compared to other coccolithophore species (i.e., Emiliania huxleyi and Gephyrocapsa oceanica). The findings demonstrate that the production of particulate organic carbon (POC) and inorganic carbon (PIC) were affected by the high pCO2 condition. Finally, it was shown that coccolithophore responses exhibit specific variations. The experimental design and statistical analysis conducted are robust, and the results hold potential interest for the readers of this journal. I should only note minor adjustments and a technical review of English grammar.
1) Abstract: It is not necessary to compare/discuss the results here. It would be interesting to demonstrate numerically the results presented in the present study.
2) Keywords: Out of the five keywords, three are already present in the title. It would be interesting to introduce alternative words here.
3) Line 115: Consider change “steady” to “stationary”
4) Line 178: Consider change “treatments” to “irradiance”
5) Some fundamental concepts regarding photoacclimation and calcification can be better explained according to the following references:
a. https://doi.org/10.1007/s00253-022-12131-6
b. https://doi.org/10.1007/s00253-022-12131-6
Minor editing of English language required
Response to Reviewer 2 Comments
Point 1: Abstract: It is not necessary to compare/discuss the results here. It would be interesting to demonstrate numerically the results presented in the present study.
Response 1: Thanks for your comment. The discussion/comparison of the results have been removed in the abstract.
Point 2: Keywords: Out of the five keywords, three are already present in the title. It would be interesting to introduce alternative words here.
Response 2: Corrections have been made in lines 36-37. “Keywords: Phytoplankton; Coccolithophores; CO2; Irradiance; Photosynthesis; Calcification, Carbon Fixation”
Point 3: Line 115: Consider change “steady” to “stationary”
Response 3: This experiment used semi-continuous cultures, in which the cultures were diluted with seawater medium daily in order to maintain the cell growth at the exponential phase. And the samples were also collected at relatively “steady” exponential growth phase. The “stationary” is generally used in butch culture, which is reached after the exponential phase, usually with nutrient limitation. Therefore, we have added some detailed information on how the semi-continuous incubation was conducted to make it clear to the readers: “The cell densities were monitored daily to indicate the growth. For the semi-continuous incubation, daily dilutions were performed using f/20 medium [17, 22] to adjust the biomass to that of the previous day, after the cells were pre-acclimated at the experimental irradiance and CO2 concentration conditions for 3 days. This allowed the cell growth kept in the exponential phase throughout the incubation [30].”.
Reference
Cai, T.; Feng, Y.; Wang, Y.; Li, T.; Wang, J.; Li, W.; Zhou, W. The Differential Responses of Coastal Diatoms to Ocean Acidification and Warming: A Comparison Between Thalassiosira sp. and Nitzschia closterium f. minutissima. Frontiers in Microbiology 2022, 13, 851149.
Point 4: Line 178: Consider change “treatments” to “irradiance”
Response 4: The suggested change has been made in line 193. “Different letters represent significant differences between different irradiance or CO2 treatments (p < 0.05).”
Point 5: Some fundamental concepts regarding photoacclimation and calcification can be better explained according to the following references:
- https://doi.org/10.1007/s00253-022-12131-6
- https://doi.org/10.1007/s00253-022-12131-6
Response 5:This reference has been cited at the appropriate place in the manuscript. Please see reference [37].
Reviewer 3 Report
Dear Authors,
I have meticulously reviewed the article ''Ocean acidification affected the response of coastal coccolithophore Pleurochrysis carterae to irradiance'' which submit to the Biology journal. While the author's research topic is intriguing and aligns with current research trends, there are numerous errors and issues that require clarification in the manuscript. I kindly request the author to address the following points.
The article's line numbers should be consecutively numbered instead of renumbering after 100.
Introduction
2. Page 3, Line 46-47: Update the citations to more recent sources, referring to the National Oceanic and Atmospheric Administration (NOAA) website.
Page 3, Line 54:"increasing" instead of "increase."
Page 3, Line 59: Move the reference [6] before the comma.
Page 3, Line 79: Clarify the meaning of "cellular POC content."
Materials and Methods
6. Page 4: Explain why the author use 500 and 800 umol photons for the experiment. Provide the type of light. What is the deviation of the irradiance?
Page 4, Line 7: Ensure proper spacing in "CO2-saturated."
Page 4, Line 23: Why the author used an organic carbon analyzer to detect inorganic carbon in water. Please explain.
Page 4, Line 25: Consider including salinity when applying CO2SYS for inorganic carbon value.
Page 4, Line 33: How to make sure that chlorophyll is fully dissolved, are algae broken by physical?
Page 5, Line 47: Add space between "USA)." and "The."
Results
12. Page 6, Line 76 & 95: Add a percentage symbol to the Growth rate and Production rate in Figure 1 and Figure 2.
Page 6, Line 95: Explain the meaning of "B&D" in Figure 2.
Page 8, Line 176: The statistical symbol in Figure 3 is wrong, in addition, the author does not show the t-test result.
Page 9, Line 213: Ensure completeness of the information provided by two-way ANOVA in Table 1.
Page 9, Line 216: Transfer Results 3.4 to the Discussion section and explain differences among the coccolithophore species. Also, clarify if the algae used by the authors are representative of this algae.
Discussion
17. Page 14, Line 406: Properly subscript "CO2" in "COâ‚‚"
Table S1: Include the standard deviation of the data in this table and provide a discussion in the Results and Discussion section.
Figure S2: Explain the symbols used in Figure S2.
Add a discussion on the effect of experimental variables on the morphology of algae.
I hope the author will carefully address these points to improve the clarity and accuracy of the manuscript. Once these issues are resolved, the article will significantly enhance its quality and contribute effectively to the current research trends.
Response to Reviewer 3 Comments
Point 1: The article's line numbers should be consecutively numbered instead of renumbering after 100.
Response 1: Thank you. We have carefully checked the line numbers and make the changes especially after line 100 for consecutively numbering.
Point 2: Page 3, Line 46-47: Update the citations to more recent sources, referring to the National Oceanic and Atmospheric Administration (NOAA) website.
Response 2: We have updated the references as following: ”Shukla, P.R.; Skea, J.; Calvo Buendia, E.; Masson-Delmotte, V.; Pörtner, H.O.; Roberts, D.; Zhai, P.; Slade, R.; Connors, S.; Van Diemen, R. IPCC, 2019: Climate Change and Land: an IPCC special report on climate change, desertification, land degradation, sustainable land management, food security, and greenhouse gas fluxes in terrestrial ecosystems. 2019.“
Point 3: Page 3, Line 54:"increasing" instead of "increase."
Response 3: Corrections have been made accordingly.
Point 4: Page 3, Line 59: Move the reference [6] before the comma.
Response 4: The suggested change has been made in line 61 in the submitted pdf file.
Point 5: Page 3, Line 79: Clarify the meaning of "cellular POC content."
Response 5: The meaning has been clarified in lines 83-84 as “cellular particulate organic carbon (POC)”.
Point 6: Page 4: Explain why the author use 500 and 800 umol photons for the experiment. Provide the type of light. What is the deviation of the irradiance?
Response 6: The relevant explanations are as follows.
- Study shows that daily irradiance has been reported to range from 77 to 740 μmol photons m−2 s−1 in shallow mixed coastal waters. So, we designed the irradiance range from 50 to 800 μmol photons m−2 s−1 and relevant content was stated in the introduction. “Daily irradiance has been reported to range from 77 to 740 μmol photons m−2 s−1 in shallow mixed coastal waters and was predicted to be higher owing to climate change [5]. Increased CO2 and irradiance bring significant challenges to marine ecosystems.”
References
Heiden, J.P.; Kai, B.; Scarlett, T. Light Intensity Modulates the Response of Two Antarctic Diatom Species to Ocean Acidification. Frontiers in Marine Science 2016, 3, 260.
- The type of light was added in lines 113. “The light source is light-emitting diodes (LED) lamps.”
- For all experiments, which were done in a thermo-controlling incubator, the irradiance at the location of each bottle was determined to be the set value, but the specific data on irradiance were not recorded.
Point 7: Page 4, Line 7: Ensure proper spacing in "CO2-saturated."
Response 7: The suggested corrections have been made in the revised manuscript.
Point 8: Page 4, Line 23: Why the author used an organic carbon analyzer to detect inorganic carbon in water. Please explain.
Response 8: This was the only equipment available in the lab at the time. Actually, the rationale of TOC analyzer (TOC-LCPH, Shimadzu, Japan) is based on CO2 detection. Without sample combustion, but only acidification, it enables determination of total inorganic carbon concentration in water samples.
Point 9: Page 4, Line 25: Consider including salinity when applying CO2SYS for inorganic carbon value.
Response 9: Yes, we included salinity for the calculation, and it has been pointed out in the text: “Seawater salinity was determined using an optical salinity meter (LS10T, Ruiming, Shanghai). Total dissolved inorganic carbon (DIC) was determined using a total organic carbon analyzer (TOC-LCPH, Shimadzu, Japan). The pH level was determined at 15 °C using a pH meter (SevenCOmpactTM S210K, Mettler Toledo, Switzerland), calibrated with NBS calibration solutions. Total alkalinity (TA) was determined by performing potentiometric titrations on filtrated samples (0.6 μm) [29]. The carbonate chemistry in the incubation system was estimated using the program CO2SYS based on TA, pH, temperature, salinity, and phosphate concentration both in the beginning, middle and final sampling of the incubation [29].”
Reference
Dickson, A.; Millero, F.J. A comparison of the equilibrium constants for the dissociation of carbonic acid in seawater media. Deep Sea Research Part A. Oceanographic Research Papers 1987, 34, 1733-1743.
Point 10: Page 4, Line 33: How to make sure that chlorophyll is fully dissolved, are algae broken by physical?
Response 10:In order to fully dissolve the chlorophyll, the acetone extraction time was more than 12 h at -20 °C and the samples were vigorously shaken in a dark before analyzing. This is also revised in lines 151-153: “Chl.-a was extracted using 90% acetone solution at -20 °C for 12 hours. The samples were vigorously shaken in dark before being analyzed using the acidification method with a fluorometer (Trilogy, Turner Designs, USA) [32]”.
Reference
Strickland, J. A practical handbook of seawater analysis 167. Bull. Fish. Res. Board Can., Ottawa 1972.
Point 11: Page 5, Line 47: Add space between "USA)." and "The."
Response 11: A space has been added.
Point 12: Page 6, Line 76 & 95: Add a percentage symbol to the Growth rate and Production rate in Figure 1 and Figure 2.
Response 12: We have made adjustment to the structure of the data and added the inhibition rate of acidification on the growth, POC and PIC production rates of coccolithophores in Figure 1, which is presented by percentage, and the revised results are more intuitive and clearer than before.
Point 13: Page 6, Line 95: Explain the meaning of "B&D" in Figure 2.
Response 13:The purpose of the figure2B and D was to highlight the changes in CO2 concentration affecting the physiological response of P. carterae to irradiance, but the presentation was unclear, so a new expression was used in figure1, using the rate of inhibition.
Point 14: Page 8, Line 176: The statistical symbol in Figure 3 is wrong, in addition, the author does not show the t-test result.
Response 14:Thank you for pointing out this problem. The significance of different irradiance or CO2 concentration treatments was determined by Tukey’s multiple comparisons test, which was not indicated in the previous paper. We added the content in lines 169-172. “The significance between different treatments was tested by Tukey’s post-hoc multiple comparisons test. The effects of CO2 concentration (LC and HC) on the saturated irradiance and the maximum values of growth, POC and PIC production rates were compared using t-test.”
Point 15: Page 9, Line 213: Ensure completeness of the information provided by two-way ANOVA in Table 1.
Response 15:The degrees of freedom have been added in the legend.
Table 2:Results of two-way ANOVAs of the effects of irradiance and CO2 and their interaction on growth rate, Chl.-a and POC, PIC, PON and POP contents, PIC/POC, C/N, C/P and N/P ratios, POC production and PIC production. Asterisk indicates significance at p<0.05 level.
|
Parameters |
|
CO2 |
|
Irradiance |
|
Irradiance×CO2 |
||||||
|
df |
F |
p |
|
df |
F |
p |
|
df |
F |
p |
||
|
Growth rate |
|
1 |
1.30 |
0.27 |
|
3 |
28.27 |
<0.01* |
|
3 |
1.97 |
0.16 |
|
Chl.-a |
1 |
2.51 |
0.13 |
3 |
103.2 |
<0.01* |
3 |
0.72 |
0.55 |
|||
|
POC |
1 |
10.69 |
0.01* |
3 |
21.55 |
<0.01* |
3 |
2.61 |
0.09 |
|||
|
PIC |
1 |
4.94 |
0.04* |
3 |
12.69 |
<0.01* |
3 |
0.16 |
0.92 |
|||
|
PON |
1 |
5.22 |
0.04* |
3 |
39.43 |
<0.01* |
3 |
1.27 |
0.32 |
|||
|
POP |
1 |
3.82 |
0.07 |
3 |
28.75 |
<0.01* |
3 |
1.65 |
0.22 |
|||
|
C/N |
1 |
0.01 |
0.93 |
3 |
10.23 |
<0.01* |
3 |
2.78 |
0.08 |
|||
|
C/P |
1 |
15.22 |
<0.01* |
3 |
8.51 |
<0.01* |
3 |
7.38 |
<0.01* |
|||
|
N/P |
1 |
20.22 |
<0.01* |
3 |
3.85 |
0.03* |
3 |
4.49 |
0.02* |
|||
|
PIC/POC |
1 |
0.05 |
0.83 |
3 |
1.88 |
0.17 |
3 |
0.20 |
0.89 |
|||
|
POC Prod |
1 |
8.26 |
0.01* |
3 |
35.32 |
<0.01* |
3 |
2.35 |
0.11 |
|||
|
PIC Prod |
1 |
2.91 |
0.11 |
3 |
27.32 |
<0.01* |
3 |
0.59 |
0.63 |
|||
Point 16: Page 9, Line 216: Transfer Results 3.4 to the Discussion section and explain differences among the coccolithophore species. Also, clarify if the algae used by the authors are representative of this algae.
Response 16:
1)We have revised this section and adjusted it to discussion (4.4): “By fitting previously published results of OA and irradiance effects on other coccolithophore species, interspecies responses were observed. The saturating irradiance for POC and PIC production rates of both E. huxleyi (calculated based on data reported Jin et al. 2017 [59]) and G. oceanica (calculated based on data in Zhang et al. 2015 [60]) were higher than those of the larger celled species P. carterae (Figure 6). Furthermore, the saturated irradiance and the corresponding growth and POC production rates of E. huxleyi increased under OA, while the response of P. carterae was rather smaller. Contrarily, those of G. oceanica decreased with increased CO2 concentration.
References
Langer, G.; Nehrke, G.; Probert, I.; Ly, J.; Ziveri, P. Strain-specific responses of Emiliania huxleyi to changing seawater carbonate chemistry. Biogeosciences 2009, 6, 2637-2646.
Jin, P.; Ding, J.; Xing, T.; Riebesell, U.; Gao, K. High levels of solar radiation offset impacts of ocean acidification on calcifying and non-calcifying strains of Emiliania huxleyi. Marine Ecology Progress 2017, 568, 47-58.
Zhang, Y.; Bach, L.T.; Schulz, K.G.; Riebesell, U. The modulating effect of light intensity on the response of the coccolithophore Gephyrocapsa oceanica to ocean acidification. Limnology and Oceanography 2015, 60, 2145-2157.
2)Emiliania huxleyi and Gephyrocapasa oceanica are the two most abundant coccolithophore species globally (Thierstein and Young, 2004), especially in the oceanic regions. And the species in our study is a representative species in the coastal regions (Liu et al., 2019).
References
Thierstein, H.R.; Young, J.R. Coccolithophores: from molecular processes to global impact. Springer Science & Business Media 2013.
Liu, H., Sun, X.,Sun, J., Thangaraj, S., Zhang, G., Li, H., An, X. Morphology, phylogenetic position, and ecophysiological features of the coccolithophore Chrysotila dentata (Prymnesiophyceae) isolated from the Bohai Sea, China. Phycologia 2019, 58, 628–639.
Point 17: Page 14, Line 406: Properly subscript "CO2" in "COâ‚‚"
Response 17: We have fixed the problem.
Point 18: Table S1: Include the standard deviation of the data in this table and provide a discussion in the Results and Discussion section.
Response 18:The standard deviations of the data have been added in the table. Descriptions of the data have been added in the section of results 3.1 “Carbonate system in experiments”. The previous Table S1 has now become Table 1 in the revised manuscript.
Table 1. The seawater carbonate chemistry on the final sampling day. pH and DIC (total inorganic carbon) were directly measured values. TA, [HCO3-], [CO32-] and CO2 were calculated using CO2SYS. Values shown are the mean ± SEM of triplicate samples. Different letters represent significant differences between different irradiance or CO2 treatments (p < 0.05).
|
Treatment |
pHNBS |
TA μmol kg-1 |
DIC μmol kg-1 |
[HCO3-] μmol kg-1 |
[CO32-] μmol kg-1 |
CO2 ppm |
|
50 μmol photons m−2·s−1 +400 ppm |
8.16 ± 0.03a |
2303 ± 8a |
2083 ± 5a |
1927 ± 11a |
140 ± 8a |
420 ± 29a |
|
50 μmol photons m−2·s−1 +800 ppm |
7.91 ± 0.01b |
2354 ± 3b |
2229 ± 2b |
2111 ± 2b |
86 ± 2bc |
812 ± 17b |
|
200 μmol photons m−2·s−1 +400 ppm |
8.17 ± 0.00a |
2299 ± 12a |
2074 ± 9a |
1916 ± 8a |
143 ± 2a |
405 ± 3a |
|
200 μmol photons m−2·s−1 +800 ppm |
7.98 ± 0.02c |
2265 ± 8a |
2135 ± 16c |
2045 ± 8cd |
77 ± 0b |
857 ± 8b |
|
500 μmol photons m−2·s−1 +400 ppm |
8.16 ± 0.02a |
2278 ± 6ab |
2062 ± 6ab |
1909 ± 11a |
137 ± 7a |
420 ± 27a |
|
500 μmol photons m−2·s−1 +800 ppm |
7.88 ± 0.01b |
2277 ± 39a |
2159 ± 11cd |
2015 ± 21c |
94 ± 8c |
685 ± 39c |
|
800 μmol photons m−2·s−1 +400 ppm |
8.14 ± 0.01a |
2239 ± 15b |
2035 ± 13b |
1889 ± 11a |
129 ± 2a |
435 ± 6a |
|
800 μmol photons m−2·s−1 +800 ppm |
7.91 ± 0.01bc |
2296 ± 7a |
2173 ± 3d |
2059 ± 2d |
84 ± 2bc |
798 ± 17b |
Point 19: Figure S2: Explain the symbols used in Figure S2
Response 19:The meanings of symbols have been added. “LC means low CO2 concentration; HC means high CO2 concentration.”
Point 19: Add a discussion on the effect of experimental variables on the morphology of algae.
Response 19: The experiments found that acidification slightly decreased the size of P. carterae (p > 0.05, Figure 2S), but we didn’t examine the morphology of P. carterae in our study, but according to the reviewer’s comment, we have added the following text to discussion the effects of ocean acidification on the morphology of coccolithophore Emiliania huxleyi:” Although our results showed that OA inhibited calcification of the P. carterae, the effect was not statistically significant. It is noteworthy that P. carterae used in our study was isolated from coastal environment, with seawater chemistry more fluctuating than that in the oceanic environment. This likely resulted in higher adaptive capacity of P. carterae to OA than E. huxleyi. Previous study suggested that the calcification process of E. huxleyi could be sensitive to OA, with malformed coccoliths observed [54].”
Reference
Ziveri, P.; Passaro, M.; Incarbona, A.; Milazzo, M.; Rodolfo-Metalpa, R.; Hall-Spencer, J.M. Decline in coccolithophore diversity and impact on coccolith morphogenesis along a natural CO2 gradient. The Biological Bulletin 2014, 226, 282-290.
Reviewer 4 Report
Your work is interesting but a lot of ammendments are necessary.
Line 17 write POC and PIC production
Line 21. write. Emiliania huxleri has received much attention and has been...
Line 23. unknown
Line 26. You write throughout the manuscript pCO2 but you refer to CO2 concentration as ppm. pCO2 is partial pressure with units mmHg, Pa, etc. So please ammend in the rest of the text this issue. Instead of pCO2 you should write CO2 concentration.
Lines 27-28. write ...POC (....) and PIC (....) production rates ...
Line 29. write. POC and PIC production.
Lines 30-31. rephrase the sentence. Its not clear what you claim. You wrote in line 29 ..under low irradiance, here you write ...higher irradiance.
Line 35. coccolithophores. the same in line 37.
Line 44. industrial revolution.
Line 46. instead of partial pressure write "concentration as I indicated in my previous comment.
Line 54 instead of "that" write "for"
Lines 73-76. Rewrite the whole paragraph as: Previous studies mainly focused on the cosmopolitan species Emiliania huxleyi ........
Line 82. I think it is better to put the reference number [30] at the end of sentence in line 84.
Line 98. write 12L:12D.
Lines 102-103. Same as before, write CO2 concentrations instead of pCO2.
There is a major issue in materials and methods. in Lines 107-108 you describe something. I cannot figure out how you managed to have 400 and 800 ppm CO2 concentrations. This is crucial. Describe in detail your method of achieving these two concentrations and additionally the instrument or method by which you measure these concentrations.
Kines 111-112. Instead of cell abundance write cell density.
Lines 112-113. What do you mean by ...within 2 hours of the light period? Rewrite the sentence.
Lines 113-114. I dont undersyand, how you maintained CO2 concentration steady? How did you measure it?
Line 115. Delete "was".
Line 116. ....yielded the total incubation ..... What is that? I dont understand. Rewrite the sentence so as to make sense.
Line 132. cell density not cell abundance.
Line 166. ...inhibited growth ....
Line 167. .....compared to μmax. Where are the data for μmax? What is the valye of μmax?
Line 169. Something is missing here, I dont understand what you mean. Rewrite please.
Line 176. Use Chl.-a.
Line 178. What do you mean by ....between other treatment? What do you compare values for every pair of treatment or values across all tratments? The notation letters for statistical differences are very confusing. I cannot figure out what you compare. The same applies and to all other Figures.
Line 183. ..the highest value of 2.86 pg .....
Line 217. What do you mean by "parallel samples"? Do you mean similar species samples? Clarify and rewrite please.
Line 218. Rewrite the sentence its confusing.
Line 220. ....with a range of change of within 31.41%. Very bad syntax and confusing. 31.41% of what?
Line 225 ....indicating that E. huxleyi....
Line 238. Again, what are parallel samples?
Line 256. changed
Line 261. Delete "that".
Line 268. was manifested
Line 271. photosynthesis not the photosynthesis.
Lines 287-288. This is the only point in your text that the term pCO2 is justified.
Line 318. reveal.
Lines 334-335. This contadiction. Which contadiction? I dont understand. Please rewrite the phrase.
Please make the necessary ammendment indicated.
Response to Reviewer 4 Comments
Point 1: Line 17 write POC and PIC production
Response 1: Corrections have been made in the manuscript.
Point 2: Line 21. write. Emiliania huxleri has received much attention and has been...
Response 2: The correction has been made in the manuscript.
Point 3: Line 23. unknown
Response 3: The correction has been made in the manuscript.
Point 4: Line 26. You write throughout the manuscript pCO2 but you refer to CO2 concentration as ppm. pCO2 is partial pressure with units mmHg, Pa, etc. So please ammend in the rest of the text this issue. Instead of pCO2 you should write CO2 concentration.
Response 4: Based on your comment, we have made the corrections throughout the whole manuscript.
Point 5: Lines 27-28. write ...POC (....) and PIC (....) production rates ...
Response 5:The correction has been made.
Point 6: Line 29. write. POC and PIC production.
Response 6: The correction has been made.
Point 7: Lines 30-31. rephrase the sentence. Its not clear what you claim. You wrote in line 29 ..under low irradiance, here you write ...higher irradiance.
Response 7: The suggested change has been made in lines 30-33. “Ocean acidification weakened the particulate organic carbon (POC) production of Pleurochrysis carterae and the inhibition rate was decreased with increasing irradiance, indicating ocean acidification may affect the tolerating capacity of photosynthesis to higher irradiance. “
Point 8: Line 35. coccolithophores. the same in line 37.
Response 8: Corrections have been made in manuscript.
Point 9: Line 44. industrial revolution.
Response 9: The correction has been made.
Point 10: Line 46. instead of partial pressure write "concentration as I indicated in my previous comment.
Response 10:The incorrect expression has been rectified.
Point 11: Line 54 instead of "that" write "for"
Response 11:The correction has been made.
Point 12: Lines 73-76. Rewrite the whole paragraph as: Previous studies mainly focused on the cosmopolitan species Emiliania huxleyi ........
Response 12:We appreciate the reviewer for this recommendation. The expression has been modified accordingly.
Point 13: Line 82. I think it is better to put the reference number [30] at the end of sentence in line 84.
Response 13:The correction has been made.
Point 14: Line 98. write 12L:12D.
Response 14:The expression has been added in the revised.
Point 15: Lines 102-103. Same as before, write CO2 concentrations instead of pCO2.
Response 14: The incorrect expression has been rectified.
Point 16: There is a major issue in materials and methods. in Lines 107-108 you describe something. I cannot figure out how you managed to have 400 and 800 ppm CO2 concentrations. This is crucial. Describe in detail your method of achieving these two concentrations and additionally the instrument or method by which you measure these concentrations.
Response 16: Based on the comments, the method of CO2 regulation has been added to the lines 118-129.” The f/20 medium was pre-aerated with filtered air/ CO2 and air mixture to achieve the CO2 concentrations in the ambient (400 ppm)/ OA (800 ppm) treatments. During the manipulation experiments, the seawater carbonate chemistry was adjusted by constant bubbling of ambient air/ CO2 and air mixture into each incubation bottle. The CO2 and air mixture were obtained using a CO2 enriching device (CW100B, Ruihua, Wuhan, China). The incubation experiment was stared with low cell density of ~104 cells mL-1 to maintain an optically thin cell density and minimize the photosynthetic effects on the carbonate chemistry in the medium and cell self-shadings. The seawater pH in each incubation bottle was measured daily within the first 2 hours of the light period. The f/20 medium used for daily dilution in the semi-continuous incubation was also pre-aerated in order to maintain relatively constant pCO2 in each experimental treatment.”
References
Feng, Y.; Chai, F.; Wells, M.L.; Liao, Y.; Li, P.; Cai, T.; Zhao, T.; Fu, F.; Hutchins, D.A. The combined effects of increased pCO2 and warming on a coastal phytoplankton assemblage: from species composition to sinking rate. Frontiers in Marine Science 2021, 8, 622319.
Yang, Y.; Gao, K. Effects of CO2 concentrations on the freshwater microalgae, Chlamydomonas reinhardtii, Chlorella pyrenoidosa and Scenedesmus obliquus (Chlorophyta). Journal of Applied Phycology 2003, 15, 379-389.
Point 17: Kines 111-112. Instead of cell abundance write cell density.
Response 17: We have made the suggested corrections throughout the whole manuscript.
Point 18: Lines 112-113. What do you mean by ...within 2 hours of the light period? Rewrite the sentence
Response 18:We have made the correction in lines 126-127.” The seawater pH in each incubation bottle was measured daily within the first 2 hours of the light period.”
Point 19: Lines 113-114. I dont understand, how you maintained CO2 concentration steady? How did you measure it?
Response 19: The parameters of the seawater carbonate system were determined as described in lines 135-143: “Seawater salinity was determined using an optical salinity meter (LS10T, Ruiming, Shanghai). Total dissolved inorganic carbon (DIC) was determined using a total organic carbon analyzer (TOC-LCPH, Shimadzu, Japan). The pH level was determined at 15 °C using a pH meter (SevenCOmpactTM S210K, Mettler Toledo, Switzerland), calibrated with NBS calibration solutions. Total alkalinity (TA) was determined by performing potentiometric titrations on filtrated samples (0.6 μm) [29]. The carbonate chemistry in the incubation system was estimated using the program CO2SYS based on TA, pH, temperature, salinity, and phosphate concentration both in the beginning, middle and final sampling of the incubation [29].”
As mentioned in point 16, we have clearly pointed out how the carbonate chemistry was kept relatively constant: “Irradiance was measured using a light sensor logger (LI-1500, LI-COR, USA). The f/20 medium was pre-aerated with filtered air/ CO2 and air mixture to achieve the CO2 concentrations in the ambient (400 ppm)/ OA (800 ppm) treatments [27]. During the manipulation experiments, the seawater carbonate chemistry was adjusted by constant bubbling of ambient air/ CO2 and air mixture into each incubation bottle. The CO2 and air mixture were obtained using a CO2 enriching device (CW100B, Ruihua, Wuhan, China). The incubation experiment was stared with low cell density of ~104 cells mL-1 to maintain an optically thin cell density and minimize the photosynthetic effects on the carbonate chemistry in the medium and cell self-shadings. The seawater pH in each incubation bottle was measured daily within the first 2 hours of the light period. The f/20 medium used for daily dilution in the semi-continuous incubation was also pre-aerated in order to maintain relatively constant pCO2 in each experimental treatment [27]. “
Reference
Dickson, A.; Millero, F.J. A comparison of the equilibrium constants for the dissociation of carbonic acid in seawater media. Deep Sea Research Part A. Oceanographic Research Papers 1987, 34, 1733-1743.
Yang, Y.; Gao, K. Effects of CO2 concentrations on the freshwater microalgae, Chlamydomonas reinhardtii, Chlorella pyrenoidosa and Scenedesmus obliquus (Chlorophyta). Journal of Applied Phycology 2003, 15, 379-389.
Point 20: Line 115. Delete "was".
Response 20:The suggested correction has been made in the manuscript.
Point 21: Line 116. ....yielded the total incubation ..... What is that? I dont understand. Rewrite the sentence so as to make sense.
Response 21: We have changed the expression in lines 130-132.” Final sampling was conducted when the steady growth phase was reached, with the variation in the growth rates was less than 10% for at least 7 generations, which the whole incubation period is ~21 days “.
Point 22: Line 132. cell density not cell abundance.
Response 22: We have made the corrections throughout the manuscript.
Point 23: Line 166. ...inhibited growth ....
Response 23: The suggested correction has been made.
Point 24: Line 167. .....compared to μmax. Where are the data for μmax? What is the valye of μmax?
Response 24: The results have been carefully revised, and inhibition rate was used to express this change in lines 201-205” OA weakened the POC production rate at low irradiance (LL = 50 μmol photons m-2 s-1, Figure 1B and E, p<0.05). The negative effect of OA on POC production was decreased with increased irradiance, and the percentages of inhibition were 44.12%, 19.20%, 15.91% and 5.88%, respectively (Figure 1E, p < 0.05). In addition, OA also slightly weakened the PIC production (Figure 1C and F, p> 0.05).
Point 25: Line 169. Something is missing here, I dont understand what you mean. Rewrite please.
Response 25: This sentence of ” The electron transport rate ("α) " decreased from 0.05 (LC) to 0.02 (HC). “was deleted.
Point 26: Line 176. Use Chl.-a.
Response 26: We have made the suggested corrections throughout manuscript.
Point 27: Line 178. What do you mean by ...between other treatment? What do you compare values for every pair of treatment or values across all tratments? The notation letters for statistical differences are very confusing. I cannot figure out what you compare. The same applies and to all other Figures.
Response 27: Yes, pair-wise comparisons were conducted using Tukey’s post-hoc multiple comparisons test with two-way ANOVA. The notation letters also suggest statistical differences between treatment. To make it clear to understand, we have made the corrections in lines 167-173.” The individual or interactive effects of CO2 concentration and irradiance on all physiological parameters were analyzed with two-way analysis of variance (ANOVA) using GraphPad Prism 8.0 software. The significance between different treatments was tested by Tukey’s post-hoc multiple comparisons test. The effects of CO2 concentration (LC and HC) on the saturated irradiance and the maximum values of growth, POC and PIC production rates were compared using t-test. All significance level was done at the p < 0.05 level. “ The use of letters in the figures/tables now has been clarified as: “Different letters represent significant differences between different irradiance or CO2 concentration treatments (p < 0.05)”.
Point 28: Line 183. ..the highest value of 2.86 pg .....
Response 28: This section has been rephrased.
Point 29: Line 217. What do you mean by "parallel samples"? Do you mean similar species samples? Clarify and rewrite please.
Response 29: This sentence has been rephrased and removed to discussion in Lines 361-362.” All data presented in the figures are the mean values of replicate samples (Jin et al., n = 3; Zhang et al., n = 4). “
Point 30: Line 218. Rewrite the sentence its confusing.
Response 30: This sentence now has been moved to discussion.
Point 31: Line 220. ....with a range of change of within 31.41%. Very bad syntax and confusing. 31.41% of what?
Response 31: This content has also been rephrased and moved to discussion: “By fitting previously published results of OA and irradiance effects on other coccolithophore species, interspecies responses were observed. The saturating irradiance for both POC and PIC production rates of both E. huxleyi (calculated based on data reported Jin et al. 2017 [59]) and G. oceanica (calculated based on data in Zhang et al. 2015 [60]) were higher than those of the larger celled species P. carterae (Figure 6). In addition, the saturated irradiance and the corresponding growth rate, POC and PIC productions of E. huxleyi increased under OA, while the response of P. carterae was rather smaller. Contrarily, those of G. oceanica decreased with increased CO2 concentration.”
Reference
Jin, P.; Ding, J.; Xing, T.; Riebesell, U.; Gao, K. High levels of solar radiation offset impacts of ocean acidification on calcifying and non-calcifying strains of Emiliania huxleyi. Marine Ecology Progress 2017, 568, 47-58.
Zhang, Y.; Bach, L.T.; Schulz, K.G.; Riebesell, U. The modulating effect of light intensity on the response of the coccolithophore Gephyrocapsa oceanica to ocean acidification. Limnology and Oceanography 2015, 60, 2145-2157.
Point 32: Line 225 ....indicating that E. huxleyi....
Response 32: The suggested correction has been made in the manuscript.
Point 33: Line 238. Again, what are parallel samples?
Response 33: We have made the correction in lines 361-363. “All used data are the mean values of replicate samples (Jin et al., n = 3; Zhang et al., n = 4).”
Point 34: Line 256. changed
Response 34: The correction has been made in the manuscript.
Point 35: Line 261. Delete "that".
Response 35: The correction has been made.
Point 36: Line 268. was manifested
Response 36: The mistake has been fixed.
Point 37: Line 271. photosynthesis not the photosynthesis.
Response 37: We have deleted “the”.
Point 38: Lines 287-288. This is the only point in your text that the term pCO2 is justified
Response 38: The term pCO2 has been previously justified: ” Increased CO2 concentration and irradiance bring significant challenges to marine ecosystems. “
Point 39: Line 318. reveal.
Response 39: We have made the correction in lines 315-318.” In contrast, previous study revealed that the POC quota of coccolithophore E. huxleyi was not significantly affected by OA across a range of irradiance (80~200 μmol photons m−2 s−1) in the nutrient-rich environment [56], suggesting species-specific responses [11].”
Reference
Zhang, Y.; Fu, F.; Hutchins, D.A.; Gao, K. Combined effects of CO2 level, light intensity, and nutrient availability on the coccolithophore Emiliania huxleyi. Hydrobiologia 2019, 842, 127-141.
Langer, G.; Nehrke, G.; Probert, I.; Ly, J.; Ziveri, P. Strain-specific responses of Emiliania huxleyi to changing seawater carbonate chemistry. Biogeosciences 2009, 6, 2637-2646.
Point 40: Lines 334-335. This contadiction. Which contadiction? I dont understand. Please rewrite the phrase.
Response 40: We have made the correction in lines 339-343. “In addition, the study on E. huxleyi selected for the comparison was based on outdoor incubation under natural solar radiation [59], while our study on P. carterae and Zhang et al. on G. oceanica [60] were both conducted in the laboratory. The difference in light source and irradiance levels may have also caused the differential responses.”
References
Jin, P.; Ding, J.; Xing, T.; Riebesell, U.; Gao, K. High levels of solar radiation offset impacts of ocean acidification on calcifying and non-calcifying strains of Emiliania huxleyi. Marine Ecology Progress 2017, 568, 47-58.
Zhang, Y.; Bach, L.T.; Schulz, K.G.; Riebesell, U. The modulating effect of light intensity on the response of the coccolithophore Gephyrocapsa oceanica to ocean acidification. Limnology and Oceanography 2015, 60, 2145-2157.